# Nanoscale interface confinement of ultrafast spin transfer torque driving non-uniform spin dynamics

Ilya Razdolski[1], Alexandr Alekhin[1], Nikita Ilin[1,2], Jan P. Meyburg[3], Vladimir Roddatis[4], Detlef Diesing[3], Uwe Bovensiepen[5] & Alexey Melnikov[1,6]

Spintronics had a widespread impact over the past decades due to transferring information by spin rather than electric currents. Its further development requires miniaturization and reduction of characteristic timescales of spin dynamics combining the sub-nanometre spatial and femtosecond temporal ranges. These demands shift the focus of interest towards the fundamental open question of the interaction of femtosecond spin current (SC) pulses with a ferromagnet (FM). The spatio-temporal properties of the impulsive spin transfer torque exerted by ultrashort SC pulses on the FM open the time domain for probing non-uniform magnetization dynamics. Here we employ laser-generated ultrashort SC pulses for driving ultrafast spin dynamics in FM and analysing its transient local source. Transverse spins injected into FM excite inhomogeneous high-frequency spin dynamics up to 0.6 THz, indicating that the perturbation of the FM magnetization is confined to 2 nm.

[1] Physical Chemistry Department, Fritz Haber Institute of Max Planck Society, Faradayweg 4-6, 14195 Berlin, Germany. [2] Moscow Technological University MIREA, Vernadsky Ave. 78, 119454 Moscow, Russia. [3] Faculty of Chemistry, University of Duisburg-Essen, Universitätsstr. 5, 45141 Essen, Germany. [4] Universität Göttingen, Institut für Materialphysik, Friedrich-Hund-Platz 1, 37077 Göttingen, Germany. [5] Faculty of Physics and Center for Nanointegration (CENIDE), University of Duisburg-Essen, Lotharstr. 1, 47057 Duisburg, Germany. [6] Institute of Physics, Martin Luther University Halle-Wittenberg, Von-Danckelmann-Platz 3, 06120 Halle, Germany. Correspondence and requests for materials should be addressed to I.R. (email: razdolski@fhi-berlin.mpg.de) or to A.M. (email: melnikov@fhi-berlin.mpg.de).

Approaching the timescales of the underlying elementary processes[1,2], spin currents (SCs) with femtosecond pulse duration[3–5] can provide valuable fundamental insights into the ultrafast spin dynamics[6–11]. In addition to manipulating the magnetization in multilayer structures[3,12,13], ultrashort SC pulses were shown to exert spin transfer torque (STT) and thus drive the coherent magnetization dynamics in semiconductor films[14] or perpendicularly coupled magnetic bilayers[15,16]. Complementing static or frequency domain studies, this time domain approach employs quasi-instantaneous driving of collective magnetic excitations by ultrashort SC pulses thus providing access to coherent spin dynamics. However, so far STT-induced coherent magnetization dynamics was limited to the homogeneous precession characterized by $k = 0$ on the picosecond timescale, similarly to ultrafast mechanisms reported earlier[17–21]. As discussed in ref. 14, the homogeneity of the system and the quasi-homogeneous laser excitation is responsible for the uniform character of magnetization dynamics. In turn, multilayer structures made of thin (few nanometre) ferromagnetic (FM) films, similar to those investigated in ref. 15, lead to the quick increase of the frequencies of the spin-wave eigenmodes and can impede spatially resolved studies of the STT-driven excitation. Aiming at understanding characteristic microscopic STT length and timescales, we address this challenge by studying spin dynamics in considerably thicker FM layers, so that both interface and bulk STT contributions[22,23] might be active. The analysis of their importance and the relevant length scales is the major aim of our work. We argue below that in our experiment, the bulk STT is damped in the vicinity of the Fe/Au interface thus emphasizing the importance of the interface STT contribution. The latter can mediate an inhomogeneous perturbation of magnetization and promote excitation of spin waves in a FM film, which extends spin dynamics into higher-frequency range. Moreover, the spatial properties of the SC-driven STT excitation can be inferred from the spectral analysis of these high-frequency spin waves.

Here we realize this approach in epitaxial Fe/Au/Fe/MgO(001) multilayers by means of optical detection of the standing spin waves in a 15-nm-thick Fe film excited via the STT mechanism (see Fig. 1). Further, we demonstrate the complex mode structure of the excited non-uniform magnetization dynamics and show

that the ultrashort laser-induced SC pulses constitute a convenient tool to excite spin waves and study the interaction of spins with a non-collinear magnetization.

## Results

**Experiment.** The concept of our experiment is illustrated in Fig. 1. A 14 fs long laser pump pulse absorbed in the FM1 layer (emitter, thickness 16 nm) results in the emission of the subpicosecond SC pulse into Au via the non-thermal spin-dependent Seebeck effect[5]. Owing to the large lifetimes of hot electrons in Au[24], their transport in a quasi-ballistic regime delivers spin angular momentum to the second ferromagnetic (FM2) layer (collector, 14 nm thick). With this spin polarization orthogonal to the collector magnetization $\mathbf{M}_C$, both reflected and transmitted electrons in the SC pulse transfer the transient angular momentum density $\boldsymbol{\mu}(t)$ to FM2 and thus exert a STT on $\mathbf{M}_C$ (ref. 22). Locally, the interaction of $\boldsymbol{\mu}(t)$ with the magnetization $\mathbf{M}_C$ is given by[1]:

$$\frac{1}{\gamma}\frac{\partial \mathbf{M}_C}{\partial t}\bigg|_{STT} = \lambda \mathbf{M}_C \times [\boldsymbol{\mu}(t) \times \mathbf{M}_C]. \qquad (1)$$

The ultrafast impulsive STT excitation triggers spin dynamics in the collector. Strong localization of the delivered perturbation in the vicinity of the Au/Fe interface is additionally corroborated by a broad distribution of the electron energies in the SC pulse[5], which ensures effective dephasing and, together with quantum decoherence, leads to a quick decay of the bulk STT contribution[22,23]. Thus, high-frequency spin waves with non-zero $\mathbf{k}$-vectors are excited along with the homogeneous precession of magnetization $\mathbf{M}_C$ (Fig. 2a). The spin-wave dispersion for a thin magnetic film (Fig. 2b) is given by[25]:

$$f(\mathbf{k}) = \gamma \sqrt{(H_{an} + Dk^2) \cdot (H_{an} + H_{dem} + Dk^2)}, \qquad (2)$$

where $H_{dem} \approx 2.1\,\text{T}$ (Fig. 2b), $\gamma \approx 28\,\text{GHz}\,\text{T}^{-1}$ is the gyromagnetic ratio, and $D_{Fe} = 280\,\text{meV}\cdot\text{A}^2$ (ref. 26). In a film of a finite thickness, only the standing waves with $k_n = \pi n/d$ are supported (Fig. 2a,b), where the zero derivative of the magnetic moment at the interfaces[25] is ensured by the low Fe/Au interface anisotropy. Rich dynamics of the time-resolved magneto-optical Kerr effect (MOKE) signals observed in our experiments (Fig. 2c) is associated with the superposition of these long-lived standing spin-wave modes.

**Data analysis.** In lieu of fitting the raw data, we performed measurements in various combinations of the magnetization directions of FM1 and FM2, while keeping them perpendicular to each other. Based on the parity rules with respect to both $\boldsymbol{\mu}$ and $\mathbf{M}_C$ (see Supplementary Note 1), we disentangle the polar and longitudinal MOKE contributions to magnetization dynamics in the FM2 layer (Fig. 2d). This procedure results in background-free data for MOKE rotation and ellipticity (Fig. 3).

It is clearly seen that the transient MOKE signals have a complex structure demonstrating oscillations at multiple frequencies. To unravel their nature, we performed the Fourier analysis of the time-resolved MOKE data. The analysis shows the presence of four frequencies (Fig. 4a) besides the fundamental excitation, which corresponds to the uniform magnetization precession ($\mathbf{k} = 0$). We argue that these frequencies indicate the excitation of the long-lived (up to 500 ps) standing spin waves in the FM2 film. The striking match between the frequencies obtained from the Fourier analysis and those calculated from the standing spin-wave dispersion illustrated in Fig. 4a verifies our explanation. We fitted a set of the exponentially decaying oscillations with the five frequencies given by equation (2) to the experimental data. The excellent quality of the fitting results corroborate our understanding of the standing spin-wave excitation (Fig. 3, solid lines), see also Supplementary Note 7 and Supplementary Fig. 6. The difference between the MOKE rotation and ellipticity data, as well as between the polar

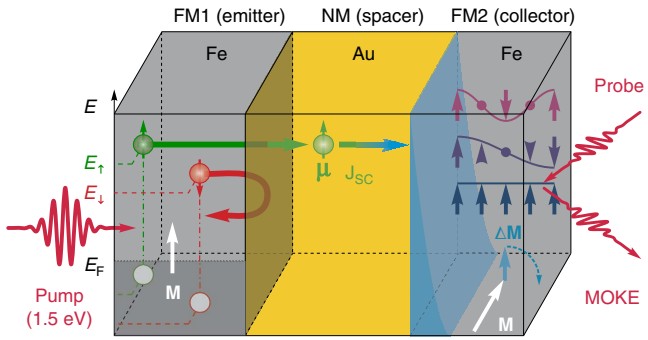

**Figure 1 | Laser-induced excitation of spin dynamics via SC pulses.** Laser pump pulse impinging on the first Fe layer (emitter) excites hot electrons at elevated energies $E_{\uparrow,\downarrow}$. Because of the unequal transmittance of the Fe/Au interface for the majority (green) and minority (red) hot electrons[5], the emission of hot electrons into Au is largely spin polarized. Having crossed the Au layer in a nearly ballistic regime, the electrons reach the second ferromagnetic (FM2) layer and transfer their spin to it. Owing to that, STT is exerted on the magnetization $\mathbf{M}_C$, which is driven out of the equilibrium and starts precessional dynamics. Due to the spatial confinement of the STT perturbation[22] (blue shaded area), spin waves with a broad spectrum of non-zero wavevectors $\mathbf{k}$ are excited and can be probed by the MOKE in the collector.

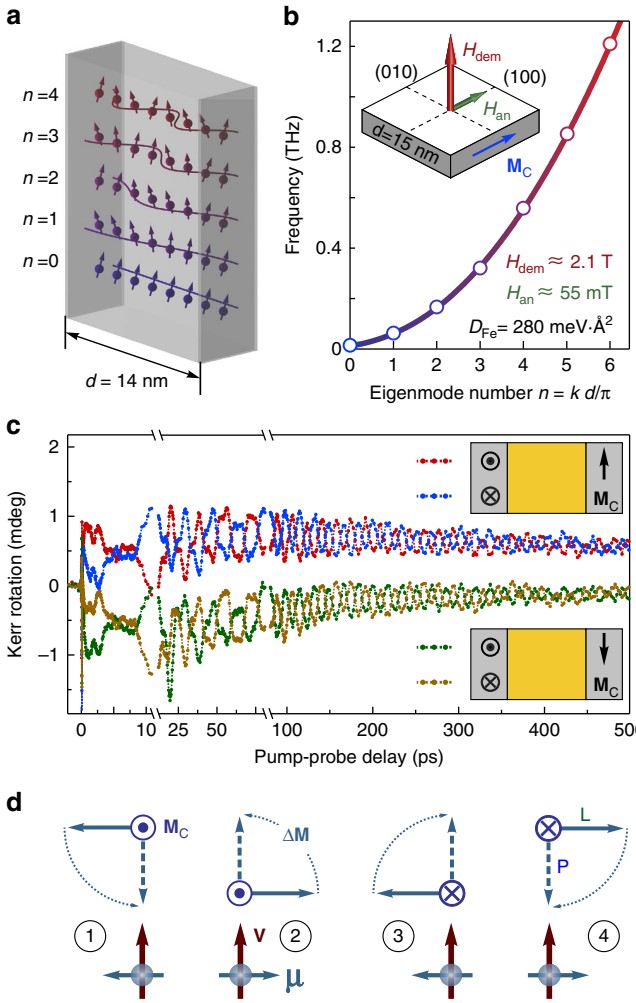

**Figure 2 | Laser-induced magnetization dynamics in the Fe/Au/Fe/MgO(001) multilayer.** (**a,b**) Spin excitations in a thin Fe(001) film magnetized in-plane. The demagnetizing field $H_{dem}$ and the two in-plane easy axes with the effective crystalline anisotropy field $H_{an}$ result in the uniform magnetization precession with a frequency $f_0 \approx 10$ GHz (ref. 29). A film with the thickness $d$ and the exchange stiffness $D_{Fe}$ can sustain standing spin waves with a discrete set of eigenwavevectors $k_n = \pi n/d$ and frequencies $f_n$ given by equation (2). (**c**) Time-resolved transient polar MOKE rotation obtained in four different geometries for orthogonal magnetizations of the emitter and collector. The periodic signals with multiple frequencies indicate the excitation of the standing spin-wave eigenmodes. (**d**) Separation of the longitudinal and polar MOKE contributions. Electrons in the SC pulse propagate with the velocity **v** and magnetic moment **μ**. For the longitudinal SC polarization **μ** and transverse magnetization of the collector $M_C$, the solid (dashed) blue arrows depict the sign of the longitudinal, L (polar, P) transient magnetization component Δ**M**. The dotted lines illustrate the appearance of the polar component of $M_C$ due to the precession of magnetization.

and longitudinal MOKE effects is attributed to their unequal sensitivity to the in-depth magnetization profile $m(z)$[27,28] and thus to various standing spin-wave eigenmodes (see Supplementary Note 2 and Supplementary Fig. 1).

## Discussion

These data unambiguously prove that the STT mechanism is capable of exciting high-frequency modes of spin precession in FM films. Note that because the data shown in Figs 2 and 3 were

obtained in the absence of an external magnetic field, the heating mechanism of the excitation of the magnetization precession relying on the ultrafast quenching of the magnetic anisotropy is inactive (see Supplementary Note 3 and Supplementary Fig. 2). To elucidate the STT-induced excitation, we now turn to the ultrafast timescale at $t < 1$ ps. Figure 4b shows the initial stage of the STT-induced magnetization dynamics, when the emitter and collector films are magnetized longitudinally and transversely, respectively. Due to angular momentum conservation, the longitudinal spin polarization of the SC pulse drives the rapid initial increase of the corresponding magnetization projection starting at ∼50 fs delay, indicating the ballistic SC propagation through the 55 nm-thick Au spacer[4]. We found the duration of the SC pulse $\tau_{SC} \lesssim 250$ fs, in agreement with the results of the direct SC measurements[5]. On the picosecond timescale, only resonant spin waves from the initially excited wavepacket with a broad distribution of **k**-vectors endure. Thus, standing spin waves in the collector are formed, giving rise to the oscillatory dynamics seen in Fig. 4b.

The transverse angular momentum is transferred from the injected electrons to the magnetization $M_C$ in the vicinity of the interface, locally driving magnetization dynamics according to equation (1). Thus, besides the SC pulse duration $\tau_{SC} \lesssim 250$ fs allowing the excitation of the modes with frequencies up to $1/2\tau_{SC} \approx 2$ THz, the spectrum of the excited magnons is limited by the **k**-vector spectrum of the delivered excitation, or, in other words, the STT characteristic depth $\lambda_{STT}$. In Fig. 4b, it is seen that the first detected oscillation is the $n = 4$ mode with a frequency of $f_4 = 0.56$ THz. As such, the $n = 5$ and the higher modes are supposedly not excited, due to either spatial or temporal limitations (see Supplementary Note 6 and Supplementary Fig. 5). The estimated $\tau_{SC}$ complies with the requirement for the efficient impulsive excitation of the $n = 5$ eigenmode, $\tau_{SC} < T_5/2 \approx 0.6$ ps. Thus, the temporal constraint can be excluded and we need to invoke the spatial inhomogeneity-driven limitation on the excited eigenmodes. Figure 4c illustrates that for the standing spin waves with open ends, the critical STT excitation depth is about a quarter of the wavelength. As such, the very fact that the eigenmode with $k_4$ is unambiguously observed in our experiments can be used for a rather conservative estimate $\lambda_{STT} \lesssim 1/4 \times 2\pi/k_4 \approx 2$ nm. We note that 2 nm is rather the upper limit of the STT depth, which can be significantly smaller. For instance, other experimental methods suggest that the STT depth in transition metals is on the order of 1–2 nm or less (see ref. 23 and references therein). Together with the L-MOKE data shown in Fig. 4b, this value indicates that the angular momentum transfer to the FM2 layer results in a 1.3° tilt of $M_C$ from its equilibrium direction within $\lambda_{STT}$.

With the ultimately short ($\lesssim 2$ nm) characteristic depth, the hot electron-induced STT remains one of the most efficient mechanisms for the excitation of the non-uniform magnetization dynamics with large **k**-vectors. For comparison, the optical penetration depth in transition metals is of the order of the skin depth $\delta \approx 10$–15 nm. In this case, the excitation of the standing spin waves with non-zero **k**-vectors would become possible in relatively thick films only, which effectively reduces the eigenmodes frequencies down to the tens of GHz (refs 29,30). In this regard, the ultrashort laser-induced pulses of SC are a unique tool capable of exciting large **k**, sub-THz spin waves in FM films. Our results demonstrate the extreme abilities of the SC pulses at exciting non-uniform spin dynamics and elucidate the interaction of the SC with a non-collinear magnetization. We found that the density of magnetic moment transferred across the Au/Fe interface per pulse is about $7 \mu_B$ nm$^{-2}$, showing high promise for the magnetization switching in thin FM layers. With these results in hand, further steps towards ultrafast spintronics can be expected in the near future.

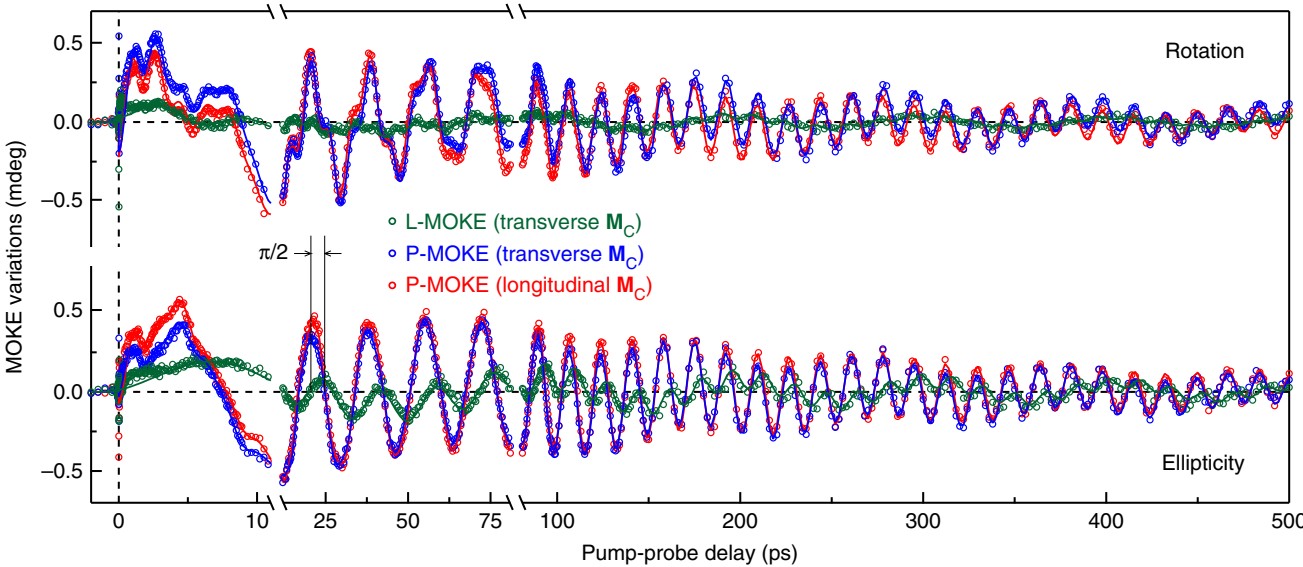

**Figure 3 | STT-induced magnetization dynamics.** Transient MOKE signals in both rotation (top) and ellipticity (bottom). Polar (P-) and longitudinal (L-) MOKE effects are separated. Solid lines are the results of the fitting procedure with a set of exponentially decaying oscillations with their frequencies given by equation (2). The pronounced similarity between the two P-MOKE traces (red and blue dots) demonstrates the robustness of the measurements. The indicated $\pi/2$ phase shift between the P- and L-MOKE components confirms the precessional nature of the magnetization dynamics.

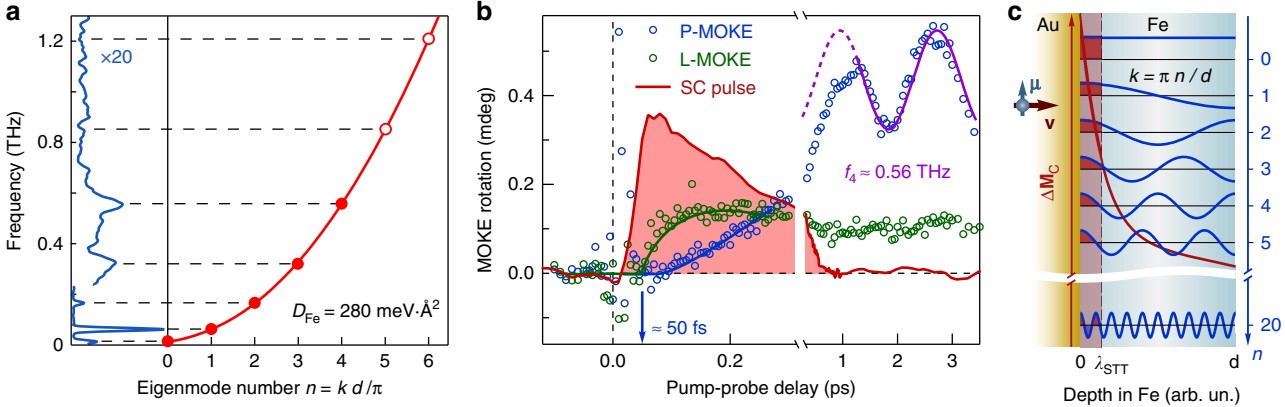

**Figure 4 | Excitation of the standing spin waves.** (**a**) Fourier spectrum of the experimental data shown in Fig. 3, averaged over several datasets (left panel). Red solid line (right panel) is the calculated spin waves dispersion curve from equation (2) with the indicated magnon stiffness $D_{Fe}$. The frequencies of the standing spin waves in a 14 nm-thick Fe film are shown in the right panel with red symbols. Along with the fundamental uniform precession ($n = 0$), four higher modes are shown which were detected in the experiment (full dots). The dashed lines illustrate the agreement between the frequencies of the Fourier peaks and those of the standing spin waves. (**b**) Transient MOKE rotation data on the ultrashort timescale. The rapid onset of the longitudinal component (green circles) is followed by a slower increase of the polar one (blue circles) due to the magnetization precession around its equilibrium. The red shaded area reproduces the SC pulse as measured in ref. 5. The purple line illustrates the 4th spin wave eigenmode with the frequency $f_4$ observed in the transient polar MOKE component. (**c**) Excitation of the higher eigenmodes is limited by the spatial in-depth profile of the STT perturbation $\mathbf{M}_C$ (red curve). The excitation region with a characteristic depth $\lambda_{STT}$ is sketched by the red shaded area. The excitation efficiency given by the convolution of the STT excitation profile and the in-depth profile of a particular standing spin-wave eigenmode (blue curves) is represented with red and blue shaded areas. In the case of a homogeneous precession (the top eigenmode with $n = 0$), this overlap is always non-zero. The eigenmodes with small non-zero $n$ are still excited albeit with smaller efficiency. However, for the eigenmode with a large $n$ (bottom, $n = 20$), the convolution yields positive and negative regions (red and blue shaded areas), which cancel each other out, thus greatly reducing the excitation efficiency.

## Methods

**Sample fabrication.** Epitaxial Fe/Au/Fe structures with 15 and 16 nm Fe layers and 55 nm Au spacer capped with 3 nm of Au were grown on optically transparent MgO(001) substrates with a thickness of 0.5 mm. The substrates were cleaned in the ultrasonics baths of ethanol, isopropanol, acetone (15 min each at elevated temperature of $\approx$310 K). After being mounted in an ultra-high vacuum chamber, the substrates were exposed to oxygen with a partial pressure of $2 \times 10^{-3}$ mbar at a temperature of 540 K for 30 min to remove spurious amounts of diamond polish left by the last fabrication step. With ultra-high vacuum conditions reached, Fe and the first nanometre of the interstitial Au layer were evaporated at 460 K. The samples were then cooled down and the additional amount of Au was evaporated at room temperature. Transparent substrate and thin capping Au layer provided optical access from both sides. The scanning transmission electron microscopy revealed excellent epitaxial quality and flatness of interfaces[5] (see Supplementary Note 4 and Supplementary Fig. 3).

**Magnetic characterization.** Before the time-resolved experiments, magnetic properties of the samples were characterized by means of the static MOKE measurements. The hysteresis loops of the MOKE rotation and ellipticity were obtained from both sides of the samples (see Supplementary Note 5 and Supplementary Fig. 4). The sample was placed in such a way that its easy anisotropy axes were oriented along the longitudinal and transverse magnetic field directions. During the measurements, a longitudinal magnetic field was scanned from $-20$ to 20 Oe, while a weak auxiliary magnetic field up to 5 Oe was applied in the transverse direction. Both fields were produced by electromagnets. This procedure allowed us to realize the two-step switching of the

magnetization from one longitudinal direction to another via the intermediate transversal direction (along the auxiliary magnetic field). In the MOKE measurements on the collector and emitter films, we obtained different width of the hysteresis loops. This allowed us to attain an orthogonal magnetic state, where the magnetizations of the collector and the emitter were perpendicular to each other.

**Time-resolved measurements.** In our back pump-front probe experiments, we used p-polarized 800 nm, 14 fs output of a commercial cavity-dumped Ti:sapphire oscillator (Mantis, Coherent) operated at 1 MHz, which was split at a power ratio 4:1 into pump and probe pulses. Both beams were chopped with different frequencies and focused with off-axis parabolic mirrors to about 10 μm spot size, resulting in a pump fluence of the order of 10 mJ cm$^{-2}$. The signal reflected from the sample was split into two identical shoulders to allow for simultaneous measurements of the MOKE rotation and ellipticity. Balanced detection scheme in each shoulder was realized with the help of a Glan-laser prism and two photodiodes. In the ellipticity shoulder, a quarter-wave plate was installed before the prism. The measurements were performed at room temperature. The zero time delay was determined before each measurement with the help of the cross-correlation second harmonic generation signal. The time-resolved MOKE data obtained on another sample with slightly different collector thickness is shown in Supplementary Fig. 6.

**Spectral data analysis.** The MOKE transients spectrum shown in the left panel in Fig. 4a was calculated using the fast Fourier transformation of the background-free data shown in Fig. 3. The low-frequency part of the spectrum (below the break) was obtained from the entire transient MOKE signals in the delay range 0–500 ps, whereas the high-frequency part (above the break) was taken from the 0–5 ps delay range only. The choice of the latter was motivated by the fact that quickly decaying high-frequency oscillations are absent at delays longer than a few tens of ps. Similar considerations apply to the spectra shown in Fig. 2c in the Supplementary Note 3. There, only the low-frequency part of the spectrum is shown.

**Calculations of density of the injected magnetic moment.** The L-MOKE data shown in Fig. 4b indicate the transient signal directly after the injection (at about 200 fs delay) of $\theta_{tr} \approx 0.14$ mdeg. Using static L-MOKE rotation $\theta_0 \approx 40$ mdeg measured in our set-up and the L-MOKE sensitivity $s(z)$ shown in Supplementary Note 2, we obtain: $\theta_0 = \int_0^d M_0 s(z) dz$, $\theta_{tr} = \int_0^\lambda \Delta M s(z) dz$, where $\lambda = 2$ nm and thus $\Delta M/M_0 \approx 0.023$, meaning 1.3 degrees tilt of magnetization. Using the static magnetic moment per Fe atom of 2.2 $\mu_B$ we get $\Delta \mu' \approx 0.042\ \mu_B$ per atom distributed within the STT depth $\lambda_{STT}$, which we assumed to be equal to 2 nm. Further, using the bcc Fe lattice constant $a = 0.287$ nm and noting that there are two atoms per unit cell in bcc lattice, we obtain total magnetic moment transferred across the interface per pulse $\Delta \mu = \Delta \mu' \times (\lambda/0.5a) \approx 0.6\ \mu_B$ per atom. From here, for the density of this transferred magnetic moment we get $\Delta \mu/a^2 \approx 7\ \mu_B$ nm$^{-2}$.

**Data availability.** The data that support the findings of this study are available from the corresponding authors on reasonable request.

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

## Acknowledgements

The authors are indebted to A. Kirilyuk, S. Sanvito, S.O. Demokritov, V.V. Kruglyak, M. Weinelt, P. Oppeneer and T. Kampfrath for fruitful discussions, as well as M. Wolf for support. This work was partially funded by the Deutsche Forschungsgemeinschaft (ME 3570/1, Sfb 616) and by the EU 7-th framework program (CRONOS).

## Author contributions

U.B. and A.M. conceived and designed the experiment. J.P.M. and D.D. fabricated the samples. V.R. performed the scanning electron microscopy of the samples. I.R., A.A., N.I. and A.M. carried out the measurements. I.R, A.A., U.B. and A.M. discussed and analysed the data. I.R. and A.M. wrote the manuscript with the help of the co-authors.

## Additional information

**Competing interests:** The authors declare no competing financial interests.

**Publisher's note**: 

