## [Peer Review File · Nature Communications]

Reviewers' comments:

Reviewer #1 (Remarks to the Author):

This is a well written manuscript on a very timely topic. The authors have successfully used 14 fs pulses onto a Fe/Au/Fe trilayer to excite a spin current pulse in one of the Fe layers, which travels through the Au to impinge on the second Fe layer (the collector) where a number of quantized perpendicular standing spin wave (PSSW) modes are excited. The experiments are very carefully carried out with great attention to detail. While I do recommend publication of the manuscript I have a number of requests that I want the authors to address:

1. It is not particularly clear why interfacial STT dominates in this experiment, when earlier experiments showed a uniform STT. It is not clear to the reader WHY this difference arises and the authors must do a better job in explaining this. I understand that the authors cite a preprint [11] where this might be better explained, but I do not find this sufficient for the present manuscript.
2. In most figures, everything is clearly labeled. However, in Fig.4b it would help the reader if the red line and red region underneath it were labeled already in the figure. Also, at first it was hard for me to understand why the SC pulse had a sudden sharp turn-off until I realized that there is a cut in the x axis and two different scales. This gap is clear in the MOKE data but less clear in the SC data since the red region is continuous. The authors should make a corresponding gap in the red region (not only the red line) to more clearly indicate the cut and change of scale.
3. In Fig.4a it is clear that the higher PSSW modes are much less excited than the first. However, in the case of $n=4$ it is actually higher than $n=3$. Would the authors venture a guess that it might be due to the spatially decaying STT excitation profile following the $n=4$ mode structure in an optimal way?
4. Finally, as there is only confinement in one dimension it is only nanoscale in the thickness direction. I recommend that the authors add the word "interface" to the title to read "Nanoscale interface confinement of...", which in my opinion makes the title describe the results more accurately.

Reviewer #2 (Remarks to the Author):

The authors Razdolski et al. report in their manuscript the observation of THz spin waves in an ultrathin Fe film excited by laser excited spin transfer torque. The spin-waves are excited by a spin-transfer torque triggered by a current pulse arising from an ultrafast laser excitation. The dynamics is probed at the backside of the sample. Standing waves are found in multiple order (up to sixth). This is very exciting for THz spintronic devices and ways to produce THz radiation via spin waves. Besides this finding the discussion of the excitation mechanism is very insightful. Such high frequency standing waves in ferromagnetic films of this thickness and thinner were only found by inelastic tunneling spectroscopy so far: All-optical methods were restricted to much smaller frequencies, e.g. thicker films. An all optical direct excitation was limited because the penetration depth is much larger and an asymmetric excitation is needed, typically 15 nm. Here this is overcome by using spin transfer torques that decays within very short length scales in the few nanometer regime estimated to about 2 nm. The authors present a very sound scientific work.

The only thing that I do not find conclusive is the estimation of the decay of the transfer torque length scale. If this could be a little bit better justified, it would be insightful for the reader. This length is derived from the vanishing of the spin wave excitation to 2 nm. These length scales seems even a little too large to me. Very helpful is the measurement of different optical contributions to the MOKE

signal. In this form I do not support a publication within Nature Communications and suggest to address the following open points.

Open questions to be addressed:

- (1) It is expected that with a viscous Gilbert damping a strong frequency dependence of the decay time should be observed (line width). Is it possible by the authors to derive the lifetime for the higher order modes following out of the LLG equation?
- (2) This would help to discuss the estimation of the characteristic depth of the spin-transfer torque. I suggest to compare the characteristic depth of 2nm estimated here to theory, where the depth is in my view much smaller. I suggest to look into calculations for example summarized in a review by Mark D. Stiles, Jacques Miltat's review in Springer Spin Dynamics in Confined Magnetic Structures III to compare with the values estimated by theory. A rigid experimental method that is more direct as previous transport experiments would be very beneficial to test theoretical descriptions.
- (3) A Bohr magneton per Fe atom transferred at the Au/Fe interface seems very high. Is there any explanation why this is so efficient, what happens to the magnetization in the layer where the moment comes from, does it demagnetize correspondingly?

Reviewer #3 (Remarks to the Author):

The paper deals with an important topic: generation of the spin dynamics of a ferromagnetic thin film by a sub-ps spin-current pulse. The structure of the experiment is not new. It is similar, for example, to the structure of experiment discussed by Schellekens et al Nat Comm 5, 4333 (2014): Two ferromagnetic metallic layers are separated by a spacer layer of nonmagnetic metal. The fs pulse impinges on the first ferromagnetic layer (FM1=emitter) generating spin-current pulse that passes through the nonmagnetic (NM) spacer and reaches the second FM layer (FM2=collector) with the magnetization orthogonal to the magnetization of FM1. The spin-current pulse initiates the spin dynamics in the FM2 layer. The time-resolved MOKE measurements are performed from the collector side to study the spin dynamics.

The important characteristic feature of the present experiment is large thickness of the FM2 layer. The spin-current pulse is absorbed close to the NM/FM2 interface that influences the time-development of the spin dynamics.

The time resolved measurements show a complex oscillating form of the MOKE signal for the time interval up to 500 ps. The authors argue that the Fourier analysis of the time resolved MOKE signals shows the presence of the 5 leading frequencies corresponding to the 5 low-energy modes of standing spin waves. The lowest frequency is related to the uniform mode that is a usual subject of previous studies of this type. Further four frequencies correspond to the nonuniform standing spin waves and demonstrate the possibility of exciting nonuniform spin-wave modes by spin-current pulses.

To prove the proposed interpretation the authors refer to a simple analytical expression $f(k)$ where f is the spin-wave frequency and k is the spin-wave wave vector. The form of the k -dependence is largely determined by the spin-stiffness parameter D taken from the neutron measurements from 1973 on bulk Fe. The k -vectors assigned to the 5 measured frequencies have the form $k_n = \pi/d * n$ ($n=0,1,2,3,4$)

where d is the thickness of the FM2 film. All 5 frequencies lie ideally on the analytical curve.

I agree with the authors that the demonstration of the possibility to excite in a controlled way THz-magnons with ultrashort spin-current pulse is an important result.

However, the authors should consider the following comments aiming at making argumentation more convincing and the physical discussion more complete.

1. I find the ideal quality with which the 5 experimental frequencies lie on the simple analytical curve rather surprising. The simple analytical dependence neglects such aspects of the problem as the inequivalence of the surface and inner atoms, the competition between collective and Stoner excitations, the dependence of the spin stiffness on temperature, sample thickness etc. Also the assignment $k_n = 2\pi/d \cdot n$ neglects the inequivalence of the surface and inner atoms and proximity to another metal. These factors complicate the shape of the excitations and make them deviate from simple sine/cosine dependences assumed by simple k_n expression.

I think it is important that the experimental results are reported for different thicknesses of the FM2 layer. Since the change of the thickness d of the FM2 layer changes the k vectors of the standing waves and their frequencies the corresponding points should lie on the same $f(k)$ curve but at different positions. Such results for FM2 films of different thickness would be a strong support for the conclusions of the paper.

2. The experimental frequencies are obtained by the Fourier analysis of the time dependences shown in Fig 3. I think it is important to provide all details of how this analysis was performed. For instance, was the whole time interval 0-500 ps used in the Fourier transformation or a part of it? Were periodical boundary conditions or an extrapolation used? The thorough description of the Fourier analysis should prove the uniqueness of the extracted frequencies or, eventually, include the discussion of their dependence on the details of the analysis.

3. Not only the frequencies but also the life times of the spin waves should be discussed as systematically as possible. Since the amplitude of the oscillations in Fig. 3 decreases with time this information is encoded in the experimental data. It is again important that mathematical side of the analysis is thoroughly described.

4. I did not find convincing the statement that the next standing wave ($n=5$) is not excited because of the small convolution with the STT pulse. From Fig. 4c the cases $n=4$ and $n=5$ do not appear principally different. It is possible that decay mechanisms, e.g., single-particle Stoner excitations, prevent the formation of the well-defined spin waves at higher energies. Or, possibly, the energies of the spin waves are shifted from the positions given by simple analytical $f(k)$ dependence. The authors briefly mention

the possible role of the short life time of the higher-energy excitations. I think this important aspect should receive more systematic attention.

5. Apparently, it follows from the left panel of Fig.4a that the excitation level of the $n=1$ mode is much higher than the excitation level of the $n=0$ mode. What is the reason for this?

On the other hand, in Supplementary Note 3 it is stated that heating induced dynamics reduces to the excitation of solely $n=0$ mode. Indeed, in Fig. 2c of Supplementary Note 3 there is no sign of the excitation of the $n=1$ mode. How this result relates to the fundamental thermodynamic property of the occupation of various excited states at non-zero temperatures?

6. At the end of the last but one paragraph of the paper the authors write about 1.3 degrees tilt of the magnetization. On the other hand, at the end of the last paragraph there is the statement that the magnetic moment transfer of about $1 \mu_B$ per Fe atom at the interface. it would be useful to make clear how these two estimations relate to each other.

In conclusion, the processes addressed in the paper: the emission, propagation and absorption of the spin-current-pulse; the initiation of the spin-dynamics in the collector and time and spatial developments of the spin dynamics are complex physical processes. The paper is an experimental one and does not contain a microscopic theoretical simulation of these processes. For future developments in this field it is very important to clearly understand the status of the conclusions drawn in the work. This understanding should be accessible to the people who are not experts in the analysis of the MOKE experiments.

RESPONSE TO REFEREES

Reviewer #1 (Remarks to the Author):

This is a well written manuscript on a very timely topic. The authors have successfully used 14 fs pulses onto a Fe/Au/Fe trilayer to excite a spin current pulse in one of the Fe layers, which travels through the Au to impinge on the second Fe layer (the collector) where a number of quantized perpendicular standing spin wave (PSSW) modes are excited. The experiments are very carefully carried out with great attention to detail. While I do recommend publication of the manuscript I have a number of requests that I want the authors to address:

1. It is not particularly clear why interfacial STT dominates in this experiment, when earlier experiments showed a uniform STT. It is not clear to the reader WHY this difference arises and the authors must do a better job in explaining this. I understand that the authors cite a preprint [11] where this might be better explained, but I do not find this sufficient for the present manuscript.

Although we are not sure, we suppose the Reviewer refers to the earlier experiments published in Refs. 15-16 in our Manuscript, where uniform magnetization dynamics rather than STT-driven excitation was reported. We have extended the introductory part of the Manuscript to improve its clarity by providing more background information as suggested by the referee: “Approaching the timescales of the underlying elementary processes, SCs with femtosecond pulse duration can provide valuable fundamental insights into the ultrafast spin dynamics. In addition to manipulating the magnetization in multilayer structures [12-14], ultrashort SC pulses were shown to exert STT and thus drive the coherent magnetization dynamics in semiconductor films [15] or perpendicularly coupled magnetic bilayers [14, 16]. Complementing static or frequency domain studies, this time domain approach employs quasi-instantaneous driving of collective magnetic excitations by ultrashort SC pulses thus providing access to coherent spin dynamics. However, so far STT-induced coherent magnetization dynamics was limited to the homogeneous precession characterized by $k = 0$ on the picosecond timescale, similarly to ultrafast mechanisms reported earlier [17-21]. As discussed in Ref. 15, the homogeneity of the system and the quasi-homogeneous laser excitation is responsible for the uniform character of magnetization dynamics. In turn, multilayer structures made of thin (few nanometer) FM films, similar to those investigated in Ref. 16, lead to the quick increase of the frequencies of the spin wave eigenmodes and can impede spatially-resolved studies of the STT-driven excitation. Aiming at understanding characteristic microscopic STT length and time scales, we address this challenge by studying spin dynamics in considerably thicker FM layers, so that both interface and bulk STT contributions [22, 29] might be active. The analysis of their importance and the relevant length scales is the major aim of our work. We argue below that in our experiment, the bulk STT is damped in the vicinity of the Fe/Au interface thus emphasizing the importance of the interface STT contribution. The latter can mediate an inhomogeneous perturbation of magnetization and promote excitation of spin waves in a FM film which extends spin dynamics into higher frequency range. Moreover, the spatial properties of the SC-driven STT excitation can be inferred from the spectral analysis of these high-frequency spin waves. In this Letter, we realize this approach

in epitaxial Fe/Au/Fe/MgO(001) multilayers by means of optical detection of the standing spin waves in a 15-nm thick Fe film excited via the STT mechanism (see Fig. 1). Further, we demonstrate the complex mode structure of the excited non-uniform magnetization dynamics and show that the ultrashort laser-induced SC pulses constitute a convenient tool to excite spin waves and study the interaction of spins with a non-collinear magnetization.”

2. In most figures, everything is clearly labeled. However, in Fig.4b it would help the reader if the red line and red region underneath it were labeled already in the figure. Also, at first it was hard for me to understand why the SC pulse had a sudden sharp turn-off until I realized that there is a cut in the x axis and two different scales. This gap is clear in the MOKE data but less clear in the SC data since the red region is continuous. The authors should make a corresponding gap in the red region (not only the red line) to more clearly indicate the cut and change of scale.

We thank Reviewer for this comment and have modified Figure 4 accordingly.

3. In Fig.4a it is clear that the higher PSSW modes are much less excited than the first. However, in the case of $n=4$ it is actually higher than $n=3$. Would the authors venture a guess that it might be due to the spatially decaying STT excitation profile following the $n=4$ mode structure in an optimal way?

Although we cannot exclude the intriguing possibility suggested by the Reviewer, we are unable to unambiguously conclude on that due to several reasons. Firstly, the detected amplitudes of the spin wave eigenmodes are governed by the MOKE sensitivity. Further, the eigenmodes excitation can be resonantly enhanced or suppressed. Moreover, the oscillating transient magnetic moment in the Au spacer might also contribute to the MOKE signals. We have added the following discussion to the Supplementary Note 2: “Similarly, in-depth MOKE sensitivity is responsible for the detected amplitudes of the excited spin wave eigenmodes in MOKE signals. We also note that the oscillating transient magnetic moment in the Au spacer might also contribute to the MOKE signals, thus making the analysis of the amplitudes non-trivial. As such, it is difficult to use MOKE amplitudes for drawing conclusions on the STT spatial profile. Furthermore, considering a simple exponentially decaying shape for the interfacial STT can be misleading as the detected amplitudes of the spin wave eigenmodes might be enhanced or suppressed by the resonances in the excitation of particular modes. In fact, this resonant behaviour is supported by the analysis of the lifetimes (and effective damping parameter) of the spin wave eigenmodes (see newly added Supplementary Note 6). It is seen there that the 4th eigenmode is very different to the other ones up to the 3rd, having a much stronger effective damping than can be expected.”

4. Finally, as there is only confinement in one dimension it is only nanoscale in the thickness direction. I recommend that the authors add the word “interface” to the title to read “Nanoscale interface confinement of...”, which in my opinion makes the title describe the results more accurately.

The title has been modified accordingly.

Reviewer #2 (Remarks to the Author):

The authors Razdolski et al. report in their manuscript the observation of THz spin waves in an ultrathin Fe film excited by laser excited spin transfer torque. The spin-waves are excited by a spin-transfer torque triggered by a current pulse arising from an ultrafast laser excitation. The dynamics is probed at the backside of the sample. Standing waves are found in multiple order (up to sixth). This is very exciting for THz spintronic devices and ways to produce THz radiation via spin waves. Besides this finding the discussion of the excitation mechanism is very insightful. Such high frequency

standing waves in ferromagnetic films of this thickness and thinner were only found by inelastic tunneling spectroscopy so far: All-optical methods were restricted to much smaller frequencies, e.g. thicker films. An all optical direct excitation was limited because the penetration depth is much larger and an asymmetric excitation is needed, typically 15 nm. Here this is overcome by using spin transfer torques that decays within very short length scales in the few nanometer regime estimated to about 2 nm. The authors present a very sound scientific work.

The only thing that I do not find conclusive is the estimation of the decay of the transfer torque length scale. If this could be a little bit better justified, it would be insightful for the reader. This length is derived from the vanishing of the spin wave excitation to 2 nm. These length scales seems even a little too large to me. Very helpful is the measurement of different optical contributions to the MOKE signal. In this form I do not support a publication within Nature Communications and suggest to address the following open points.

Open questions to be addressed: (1) It is expected that with a viscous Gilbert damping a strong frequency dependence of the decay time should be observed (line width). Is it possible by the authors to derive the lifetime for the higher order modes following out of the LLG equation?

It is indeed possible and, according to the suggestion of the Reviewer, we have added a Supplementary Note 6 where we analyze the lifetimes of the observed eigenmodes. We observe a continuous decrease of the lifetime τ (or increase of the decay rate τ^{-1}) of the spin wave eigenmodes (see Fig. 5a in the Supplementary Information). In Fig. 5b it is seen that the recalculated effective Gilbert damping remains approximately constant for the eigenmodes up to the 3-rd whereas the 4-th mode has much shorter lifetime.

(2) This would help to discuss the estimation of the characteristic depth the spin-transfer torque. I suggest to compare the characteristic depth of 2nm estimated here to theory, where the depth is in my view much smaller. I suggest to look into calculations for example summarized in a review by Mark D. Stiles, Jacques Miltat's review in Springer Spin Dynamics in Confined Magnetic Structures III to compare with the values estimated by theory. A rigid experimental method that is more direct as previous transport experiments would be very beneficial to test theoretical descriptions.

We thank the Reviewer for this comment. In the previous version we might have complicated the discussion in an unnecessary manner. In order to make our point clear, in the revised version we have moved the discussion of the short lifetime of the 5-th spin wave eigenmode to Supplementary Information and added a reference [29] to the review paper recommended by the Reviewer. The manuscript now says: "Figure 4,c illustrates that for the standing spin waves with open ends, the critical STT excitation depth is about a quarter of the wavelength. As such, the very fact that the eigenmode with k_4 is unambiguously observed in our experiments can be used for a rather conservative estimate $\lambda_{\text{STT}} < 2$ nm. We note that 2 nm is rather the upper limit of the STT depth which can be significantly smaller. For instance, other experimental methods suggest that the STT depth in transition metals is on the order of 1-2 nm or less (see [29] and references therein)".

(3) A Bohr magneton per Fe atom transferred at the Au/Fe interface seems very high. Is there any explanation why this is so efficient, what happens to the magnetization in the layer where the moment comes from, does it demagnetize correspondingly?

We are grateful to the Reviewer for this comment and modified this part of the paper to avoid confusion. The manuscript now gives a value of $7 \mu_B/\text{nm}^2$ for the density of the magnetic moment

transferred across the interface per pulse. We have also added a detailed description of the calculations leading to this estimate to the Methods section. The results of our complementary experiments allow us to conclude that the emitter layer is accordingly demagnetized, as suggested by the Reviewer. However, preliminary results indicate that this demagnetization occurs within an emission depth on the order of 5-10 nm, considerably larger than the STT depth. As such, the transport-induced demagnetization in the emitter will be accordingly weaker than the STT effect in the interfacial region of the collector.

Reviewer #3 (Remarks to the Author):

The paper deals with an important topic: generation of the spin dynamics of a ferromagnetic thin film by a sub-ps spin-current pulse. The structure of the experiment is not new. It is similar, for example, to the structure of experiment discussed by Schellekens et al Nat Comm 5, 4333 (2014): Two ferromagnetic metallic layers are separated by a spacer layer of nonmagnetic metal. The fs pulse impinges on the first ferromagnetic layer (FM1=emitter) generating spin-current pulse that passes through the nonmagnetic (NM) spacer and reaches the second FM layer (FM2=collector) with the magnetization orthogonal to the magnetization of FM1. The spin-current pulse initiates the spin dynamics in the FM2 layer. The time-resolved MOKE measurements is performed from the collector side to study the spin dynamics.

The important characteristic feature of the present experiment is large thickness of the FM2 layer. The spin-current pulse is absorbed close to the NM/FM2 interface that influences the time-development of the spin dynamics.

The time resolved measurements show a complex oscillating form of the MOKE signal for the time interval up to 500 ps. The authors argue that the Fourier analysis of the time resolved MOKE signals shows the presence of the 5 leading frequencies corresponding to the 5 low-energy modes of standing spin waves. The lowest frequency is related to the uniform mode that is a usual subject of previous studies of this type. Further four frequencies correspond to the nonuniform standing spin waves and demonstrate the possibility of exciting nonuniform spin-wave modes by spin-current pulses.

To prove the proposed interpretation the authors refer to a simple analytical expression $f(k)$ where f is the spin-wave frequency and k is the spin-wave wave vector. The form of the k -dependence is largely determined by the spin-stiffness parameter D taken from the neutron measurements from 1973 on bulk Fe. The k -vectors assigned to the 5 measured frequencies have the form $k_n = \pi/d * n$ ($n=0,1,2,3,4$) where d is the thickness of the FM2 film. All 5 frequencies lie ideally on the analytical curve.

I agree with the authors that the demonstration of the possibility to excite in a controlled way THz-magnons with ultrashort spin-current pulse is an important result. However, the authors should consider the following comments aiming at making argumentation more convincing and the physical discussion more complete.

1. I find the ideal quality with which the 5 experimental frequencies lie on the simple analytical curve rather surprising. The simple analytical dependence neglects such aspects of the problem as the inequivalence of the surface and inner atoms, the competition between collective and Stoner excitations, the dependence of the spin stiffness on temperature, sample thickness etc. Also the assignment $k_n = 2\pi/d * n$ neglects the inequivalence of the surface and inner atoms and

proximity to another metal. These factors complicate the shape of the excitations and make them deviate from simple sine/cosine dependences assumed by simple k_n expression.

I think it is important that the experimental results are reported for different thicknesses of the FM2 layer. Since the change of the thickness d of the FM2 layer changes the k vectors of the standing waves and their frequencies the corresponding points should lie on the same $f(k)$ curve but at different positions. Such results for FM2 films of different thickness would be a strong support for the conclusions of the paper.

We thank the Reviewer for this detailed comment. The caption of Fig. 4 has been modified in order to make the fit procedure more transparent. Here, we compare the peaks in the Fourier spectrum (left) with the calculated frequencies of the spin wave eigenmodes depicted with red dots. The degree of agreement between the two sets of frequencies reached within the fit procedure is illustrated with the dashed lines and is limited by the width of the Fourier peaks. We further note that low surface anisotropy of the Fe/Au interface allows for the simple analysis presented in the manuscript. Stoner excitations playing an important role at the energies of about a half of the exchange splitting are a rather unlikely candidate as the frequencies considered here do not exceed $1 \text{ THz} \approx 4 \text{ meV}$, much smaller than the Stoner gap. The dispersions (frequencies and wavevectors) of other spin wave excitations such as Damon-Eshbach etc. discussed in Ref. 27 do not fit to the experimental data, leaving no doubts about the character of the spin wave eigenmodes. The influence of the sample thickness discussed in Ref. 7 in the Supplementary Information cannot be excluded, but relatively large thickness of the collector suggests the negligibility of surface contributions and corresponding variations of the magnon stiffness D . Nevertheless, the consequences of the possible variations of the stiffness are discussed in Supplementary Note 4.

Although a systematic thickness-dependent study is beyond the scope of the present paper, we performed similar measurements on a sample with a slightly thicker collector (13.2 nm vs 12.7 nm, according to the transmission electron microscopy data) and obtained equally good agreement of the calculated and experimentally detected frequencies. We have added a Supplementary Note 7 where the corresponding data are shown. There, instead of taking the Fourier spectra we simply fitted a set of five exponentially decaying harmonics to the data and compared the obtained frequencies to the calculated ones.

2. The experimental frequencies are obtained by the Fourier analysis of the time dependences shown in Fig 3. I think it is important to provide all details of how this analysis was performed. For instance, was the whole time interval 0-500 ps used in the Fourier transformation or a part of it? Were periodical boundary conditions or an extrapolation used? The thorough description of the Fourier analysis should prove the uniqueness of the extracted frequencies or, eventually, include the discussion of their dependence on the details of the analysis.

We have added the detailed explanation of retrieving the frequencies to the Methods section of the Manuscript.

3. Not only the frequencies but also the life times of the spin waves should be discussed as systematically as possible. Since the amplitude of the oscillations in Fig. 3 decreases with time this information is encoded in the experimental data. It is again important that mathematical side of the analysis is thoroughly described.

We welcome the suggestion of the referee and added a Supplementary Note 6 where we analyze the lifetimes of the observed eigenmodes. We observe a continuous decrease of the lifetime τ (or increase of the decay rate τ^{-1}) of the spin wave eigenmodes (see Fig. 5a in the Supplementary Information). In Fig. 5b it is seen that the recalculated effective Gilbert damping parameter α remains approximately constant for the eigenmodes up to the 3-rd whereas the 4-th mode has much shorter lifetime.

4. I did not find convincing the statement that the next standing wave ($n=5$) is not excited because of the small convolution with the STT pulse. From Fig. 4c the cases $n=4$ and $n=5$ do not appear principally different. It is possible that decay mechanisms, e.g., single-particle Stoner excitations, prevent the formation of the well-defined spin waves at higher energies. Or, possibly, the energies of the spin waves are shifted from the positions given by simple analytical $f(k)$ dependence. The authors briefly mention the possible role of the short life time of the higher-energy excitations. I think this important aspect should receive more systematic attention.

We thank the Reviewer for this comment. Although Stoner excitations play an important role at the energies of about a half of the exchange splitting, they are a rather unlikely candidate to explain the absence of the $n=5$ mode because the frequencies considered here do not exceed $1 \text{ THz} \approx 4 \text{ meV}$ which is much smaller than the Stoner gap. However, there indeed might be other mechanisms impeding excitation of the higher modes such as, for example, phonon-magnon or nonlinear spin wave scattering. In order to clarify our point in the manuscript we have removed the discussion of the short lifetime of the 5-th spin wave eigenmode. The paper now says: “Figure 4,c illustrates that for the standing spin waves with open ends, the critical STT excitation depth is about a quarter of the wavelength. As such, the very fact that the eigenmode with k_4 is unambiguously observed in our experiments can be used for a rather conservative estimate $\lambda_{\text{STT}} < 2 \text{ nm}$. We note that 2 nm is rather the upper limit of the STT depth which can be significantly smaller. For instance, other experimental methods suggest that the STT depth in transition metals is on the order of $1\text{-}2 \text{ nm}$ or less (see [29] and references therein).”

5. Apparently, it follows from the left panel of Fig.4a that the excitation level of the $n=1$ mode is much higher than the excitation level of the $n=0$ mode. What is the reason for this?

We believe this has to do with the in-depth sensitivity of MOKE which governs the detected amplitudes of the spin wave eigenmodes. Also, the possible resonant enhancement or suppression of the excitation of particular eigenmodes might play an important role here. We have added the following discussion to the Supplementary Note 2: “Similarly, in-depth MOKE sensitivity is responsible for the detected amplitudes of the excited spin wave eigenmodes in MOKE signals. However, considering a simple exponentially decaying shape for the interfacial STT can be misleading as the detected amplitudes of the spin wave eigenmodes might be enhanced or suppressed by the resonances in the excitation of particular modes. In fact, this resonant behaviour is supported by the analysis of the lifetimes (and effective damping parameter) of the spin wave eigenmodes (see newly added Supplementary Note 6). It is seen there that the 4th eigenmode is very different to the other ones up to the 3rd, having a much stronger effective damping than can be expected. Furthermore, we note that the oscillating transient magnetic moment in the Au spacer might also contribute to the MOKE signals, thus making the analysis of the amplitudes non-trivial. As such, it is difficult to use MOKE amplitudes for drawing conclusions on the STT spatial profile.”

On the other hand, in Supplementary Note 3 it is stated that heating induced dynamics reduces to the excitation of solely $n=0$ mode. Indeed, in Fig. 2c of Supplementary Note 3 there is no sign of the

excitation of the n=1 mode. How this result relates to the fundamental thermodynamic property of the occupation of various excited states at non-zero temperatures?

The Reviewer is correct saying that in a thermally equilibrated system, thermodynamical considerations would suggest the excitation of the higher spin wave eigenmodes at elevated temperatures. However, these spin waves are incoherently excited with random phases and thus their mutual interference results in a mere demagnetization of the collector instead of coherent magnetization dynamics. On the contrary, the coherent spin wave eigenmodes considered in our work constitute a spin system very far from thermal equilibrium, which is why we believe that a thermodynamical description of such a system is misleading.

6. At the end of the last but one paragraph of the paper the authors write about 1.3 degrees tilt of the magnetization. On the other hand, at the end of the last paragraph there is the statement that the magnetic moment transfer of about $1 \mu_B$ per Fe atom at the interface. it would be useful to make clear how these two estimations relate to each other.

In conclusion, the processes addressed in the paper: the emission, propagation and absorption of the spin-current-pulse; the initiation of the spin-dynamics in the collector and time and spatial developments of the spin dynamics are complex physical processes. The paper is an experimental one and does not contain a microscopic theoretical simulation of these processes. For future developments in this field it is very important to clearly understand the status of the conclusions drawn in the work. This understanding should be accessible to the people who are not experts in the analysis of the MOKE experiments.

We are grateful to the Reviewer for this comment and modified this part of the paper to avoid confusion. The manuscript now gives a value of $7 \mu_B/\text{nm}^2$ for the density of transferred magnetic moment across the interface per pulse. We have also added a detailed description of the calculations leading to this estimate to the Methods section: “The L-MOKE data shown in Fig. 4b indicates the transient signal directly after the injection (at about 200 fs delay) of $\theta_{tr} \approx 0.14$ mdeg. Using static L-MOKE rotation $\theta_0 \approx 40$ mdeg measured in our setup and the L-MOKE sensitivity $s(z)$ shown in Supplementary Note 2, we obtain: $\theta_0 = \int^d M_0 s(z) dz$, $\theta_{tr} = \int^\lambda M s(z) dz$, where $\lambda=2$ nm and thus $\Delta M/M_0 \approx 0.023$, meaning 1.3 degrees tilt of magnetization. Using the static magnetic moment per Fe atom of $2.2 \mu_B$ we get $\Delta\mu' \approx 0.042 \mu_B$ per atom distributed within the STT depth λ which we assumed to be equal to 2 nm. Further, using the bcc Fe lattice constant $a=0.287$ nm and noting that there are two atoms per unit cell in bcc lattice we obtain total magnetic moment transferred across the interface per pulse $\Delta\mu = \Delta\mu' \times (\lambda/0.5a) \approx 0.6 \mu_B$ per atom. From here, for the density of this transferred magnetic moment we get $\Delta\mu / a^2 \approx 7 \mu_B/\text{nm}^2$.”

REVIEWERS' COMMENTS:

Reviewer #1 (Remarks to the Author):

The authors have addressed all my concerns and I recommend publication of the manuscript in Nature Communications.

Reviewer #2 (Remarks to the Author):

The authors have responded to the comments of the referees in a very detailed manner. This is excellent work and I strongly support the publication of the revised version in this form.

Reviewer #3 (Remarks to the Author):

I recommend the publication of the revised version of the paper.

Reviewer #1 (Remarks to the Author):

The authors have addressed all my concerns and I recommend publication of the manuscript in Nature Communications.

Reviewer #2 (Remarks to the Author):

The authors have responded to the comments of the referees in a very detailed manner. This is excellent work and I strongly support the publication of the revised version in this form.

Reviewer #3 (Remarks to the Author):

I recommend the publication of the revised version of the paper.